# Bacteria in Normal Canine Milk Analyzed by Blood Agar Medium

**DOI:** 10.3390/ani13132206

**Published:** 2023-07-05

**Authors:** Sabina Sibcic Kolasinac, Lars Moe, Vibeke Rootwelt, Henning Sørum

**Affiliations:** 1Department of Companion Animal Clinical Sciences, Faculty of Veterinary Medicine, Norwegian University of Life Sciences, Elizabeth Stephansens vei 15, 1433 Ås, Norway; sabina.sibcic@nmbu.no (S.S.K.); lars.moe@nmbu.no (L.M.); vibeke.rootwelt@nmbu.no (V.R.); 2Department of Paraclinical Sciences, Faculty of Veterinary Medicine, Norwegian University of Life Sciences, Elizabeth Stephansens vei 15, 1433 Ås, Norway

**Keywords:** culturable microbiome, dog, lactation, mammary gland, canine milk bacteriome

## Abstract

**Simple Summary:**

Albeit being crucial for survival of offspring, milk in the canine species is relatively poorly investigated. Normal milk microbiota is a proven and important source of bacteria able to colonize the gut of newborns described in several species, including humans. Scientists now consistently find a range of bacterial species in milk from healthy individuals lacking any clinical signs of mastitis. To elucidate this matter, we analyzed milk of healthy dams who delivered healthy puppies naturally and investigated whether it contains bacteria using standard culture media for aerobic and anaerobic growth of microorganisms. Results from 210 samples collected twice with an interval of 7 days in the early postpartum period reported *Staphylococcaceae* and *Enterococcaceae* to be most represented. No significant difference in bacterial growth from milk sampled first or second time was noted. Every dam had bacteria in most mammary glands, and only 14% of total samples had no bacterial growth. The composition of bacteria from all milk samples did not differ significantly, indicating consistent occurrence of bacteria in normal, healthy milk, not mere contamination. Considering the global threat posed by antibiotic resistance, their application should be avoided in individuals without additional clinical signs of inflammation.

**Abstract:**

Studies of microbiota in normal canine milk from healthy dams are sparse. As is the case with blood and urine, it was considered that milk contains no microbiota. Any discovery of bacteria in canine milk is, therefore, often noted to be a result of contamination during sampling or interpreted as mastitis and treated with antibiotics. Milk was collected twice within 19 days after natural parturition from 11 lactating dams, with no general or local clinical signs of mastitis or other disease. The skin and teats were prepared with an antimicrobial protocol prior to each milk sampling. In total, 210 milk samples were collected and assessed for a number of bacterial colonies grown on each plate. Bacterial growth was detected in 180 samples (86%). *Staphylococcus pseudintermedius*, *Enterococcus* spp., *Clostridium* spp., Coagulase-Negative Staphylococci (CoNS), *Streptococcus* spp., *Streptococcus canis*, *Bacillus* spp., *Pasteurella* spp., and *Escherichia coli* were identified from pure and/or mixed bacterial growth, listed in descending order of occurrence. Despite the small sample size, the consistent occurrence of bacteria in early postpartum dams indicates a genuine occurrence of bacteria in canine milk, rather than random contamination. The finding of bacteria in the milk of dams should not, therefore, be the sole argument for the diagnosis of mastitis.

## 1. Introduction

Milk is the perfect source of nutrients for neonatal mammals. The first milk, colostrum, secreted during the first 24–48 hours (h) of life, is abundant in immunoglobulins that help protect the offspring from infections before they have produced their own specific antibodies [1,2]. Normal milk was considered sterile for decades [3,4]. It has now, however, been reported that milk is an important source of bacteria from mothers that may colonize the gut of newborns in several species [5,6,7,8]. Different methods have been used to describe the normal milk microbiota in humans and animals [4,6,7], the role of the intestinal microbiota, and possible mechanisms by which the maternal intestinal microbiota might be transferred to and established in the offspring [3,5,7,8]. Lactic acid bacteria found in milk may act as a probiotic and have beneficial effects on the health status of both mother and neonate [9]. This healthy microbiota has antimicrobial activities such as the production of bacteriocins or other antimicrobial products like organic acids and may reduce the number of pathogenic bacteria in the gut [10]. The canine milk microbiota, however, has been only sparsely described in canine theriogenology [11].

Some papers, however, have published the link between bacteria-caused mastitis in the dog and septicemia or neonatal death in the offspring [12,13,14,15]. Poor milk quality is speculated to be one cause of fading puppy syndrome, but the evidence is unclear [16,17]. In two reports, the authors suggested that bacteria found in the milk of otherwise healthy dams could contain pathogens such as *Staphylococcus pseudintermedius* that could be transmitted from mother to neonate, causing septicemia in puppies [14,18]. The prevailing view has been that the bacterial vaginal flora of the dam and the skin surface of the teat, in addition to the microbiota of the environment, are the most important sources of the intestinal microbiota of the puppy [19]. As it has been already described in humans, it is possible that the normal canine milk microbiota also exists and may play an important role in establishing the intestinal flora of the puppies.

The aim of our study was to investigate whether there is a normal, culturable bacterial microflora in the milk of healthy dams during early lactation that causes no health issues to the puppies, and if so, to describe the bacterial species that are present and to estimate the populations of bacteria in milk using standard culture-dependent techniques for detection of bacteria.

## 2. Materials and Methods

### 2.1. Study Design

This study was designed as a prospective, observational study of milk bacteria in healthy dams. Samples were collected twice with the interval of 7 days (d), exceeding no more than 19 days following natural parturition. The hypothesis was that normal canine milk does not contain any culturable bacteria. The inclusion criteria were healthy dams of any age and breed, weight of more than 8 kg, no clinical signs of illness nor history of antibiotic treatment during the last two weeks prior to sampling and natural parturition.

All the dogs were living in private housing at a maximum of 40 km distance from the laboratory to ensure a limited time delay from sampling to further processing in the laboratory. All owners were informed about the purpose of the project and the sampling methods. A written signed consent was obtained prior to enrolment. After the second sampling, the health status of dams and puppies was followed until weaning at about 8 weeks of age.

### 2.2. Study Population

Eleven privately owned dams, primiparous and multiparous, ranging in age from two to seven years (yrs) were included and sampled twice within 19 days of parturition. The total number of milk samples collected was 210 (11 dams, with every functional mammary complex sampled twice). Because there are many orifices on the teat each of which is connected to glandular segments (compartments) within each mammary complex, we chose to use the term mammary complex to indicate that the term “milk sample” used here means the pooled milk collected from one or more ducts feeding into the apex of a single teat.

### 2.3. Milk Sampling Procedure

To minimize stress while sampling and to prevent the puppies from emptying the mammary complexes, owners were requested to separate the dam from the puppies one h prior to sampling. The milk samples were collected separately from each teat with a sterile pipette, after thorough antiseptic preparation. To reduce the risk of contamination of the milk from skin microflora, the hairs around the teats were removed by clipping before sampling and residual loose hairs were removed using a lint roller. The dam was placed on a pre-set table, covered with a disposable absorbent bed sheet, and lightly restrained as needed by the owner. All teats were disinfected using sterile compresses soaked in 70% ethanol to ensure that no hair, milk or loose epithelial cells were left on the skin of the teat apex. After evaporation of the alcohol until the skin completely dried (approximately 30 s), the first few milk droplets were discarded, and every teat was disinfected once again following the same procedure before collecting the final sample from the apex of the teat with a sterile pipette tip. Sterile disposable gloves were worn on both hands during disinfection and changed prior to sampling.

When necessary, oxytocin (Vetocin^®^ Bela-Pharm 10 IU/mL) was administered subcutaneously, twice maximum, within the span of 1 h, without exceeding a total dose of 0.2 mL per dam. Each sterile pipette tip containing a milk sample was ejected into its own small, sterile plastic bag with a zipper. All plastic bags from one sampled animal were then put into a larger plastic bag also with a zipper. The larger plastic bag was then immediately placed on ice and kept cool during transport to the laboratory for agar plating. The bacterial agar inoculation and cultivation was performed within three h of sampling.

### 2.4. Inoculation of Blood Agar Plates

The traditional agar plate culture method was employed. Aliquots of 10 µL of milk from each teat were applied onto one half of agar plates with blood agar base II (Difco) supplied with 5% cattle blood and distributed with a sterile disposable 1 µL plastic loop (Copan Italia S.p.A, Brescia, Italy, www.vwr.com). The plates were incubated for 24 h at 37 °C, in 5% CO_2_ for aerobic conditions, and for 48 h for anaerobic conditions (Mikrobiologie Anaerocult^®^ A, Merck KGaA, Darmstadt, Germany). Every milk sample collected (*n* = 210) was inoculated. The colonies of bacteria were phenotypically identified by experienced microbiologists based on the pattern of colony morphology, hemolysis, Gram staining and the results of biochemical tests using API^®^ kits (bioMérieux SA, Marcy-l’Étoile, France, https://www.biomerieux-diagnostics.com/nordic-countries (accessed on 28 September 2020) such as API Staph, API Strep, API 20 NE, API Coryne and API 20 A.

If one or more bacterial species were detected, the milk sample was defined as positive. Otherwise, it was defined as negative. The number of bacterial colonies on the inoculated agar plates was counted manually. All plates were photographed, and marking and counting of colonies were controlled using the Paint App (iPhone 10, iOS11, Standard Camera app, Apple Inc., Cupertino, CA, USA). The number of colonies for each identified bacterial species was defined as colony-forming units per mL of milk (cfu/mL) per teat and dam.

### 2.5. MALDI-TOF and 16S rDNA Sequence

A selection of 10 representative bacterial isolates from two dogs tested in a pilot analysis selected with the same criteria as for the 11 dogs included in this study was prepared for Matrix-assisted laser desorption ionization–time of flight (MALDI-TOF) identification performed with a Vitek MS spectrophotometer using the routine in vitro diagnostic database [20]. All strains were spotted twice. PCR was performed using the primers V1 (5′-AGA GTT TGA TCA TGG CTC AGA-3′) and V3 (5′-GGT TAC CTT GTT ACG ACT TC-3′). All PCR tests were performed on a 2720 Thermal Cycler (Applied Biosystem, Courtaboeuf, France) using a GoTaq Flexi DNA polymerase kit (Promega, Madison, WI, USA). High stringency conditions were applied as follows. After 5 min at 94 °C, 30 cycles were performed, which consisted of 1 min denaturation at 94 °C, 1 min hybridization at 55 °C, and 1 min elongation at 72 °C. For low stringency PCR, the hybridization step was performed at 50 °C for 1 min.

### 2.6. Statistical Analyses

All the data were recorded and analyzed using Excel and JMP^®^ (Version 14. SAS Institute Inc., Cary, NC, USA, 1989–2019) and grouped according to the age of the dam, litter size, number of previous litters and days after parturition. The milk samples were collected one week apart. The first sample from each dam was classified as week 1 (1–12 d postpartum) and the second sample as week 2 (7–19 d postpartum). A presence of bacterial colonies on blood agar plates was considered a positive sample, whereas its absence was considered as a negative sample. The number of cfu/mL was calculated as the sum, mean and standard error of the mean (SEM) for all positive bacteria identified, for all teats and per dam. The same statistics were calculated for the dams combined. Separate statistics were calculated for the isolated bacteria for all teats and dams. Using the same statistical tests, we investigated cfu/mL, per bacterial species, per week, and per right and left side and other combinations. Differences in number of cfu/mL and bacterial species among milk samples from each dam and mammary gland complex were calculated using the Wilcoxon signed ranks test. A possible relationship between individual samples from each dam and number of cfu/mL was investigated using the Kruskal–Wallis test (with the number of cfu/mL as the independent variable). Correlation analyses between mean number of cfu/mL from all teats per dam vs. age of the dam, days after parturition, litter size, bacterial species and number of previous litters were determined using bivariate analyses. Differences between groups were considered statistically significant if *p* < 0.05.

## 3. Results

### 3.1. Bacteria in Normal Milk

Of 210 milk samples, 180 (85.7%) samples were positive and 30 (14.3%) samples were negative for bacterial growth. The number of positive samples for the individual dams during the first and second sampling procedure after birth are shown in Figure 1a. Not all 10 mammary gland complexes produced milk, but we found bacterial growth from milk-producing teats. In the first sampling procedure (up to 12 days postpartum), 104 samples were collected, of which 86 (82.7%) were positive for bacterial growth and 18 (17.3%) were negative. In the second sampling procedure (up to 19 days postpartum), 106 samples were collected, 94 (88.7%) of which were positive for bacterial growth and 12 (11.3%) negative. Follow-up monitoring of the health of the dams and their puppies revealed no clinical signs concerning the health of the puppies before they were weaned at week 8 postpartum, but one mastitis case in dog no. 11 located within one mammary complex in week four postpartum.

In Figure 1b, milk samples with and without growth of bacteria were grouped into pairs of mammary glands, from I to V, where number I represented the most cranial pair of glands. There was no significant difference in bacterial growth and no growth, between the mammary gland complexes (from I to V), or between the right and left side. There was a tendency for fewer positive samples in the cranial pair of glands compared to the other four pairs, 76% vs. 85–89%, respectively. In the first pair of glands, more cases of failure of the gland (no milk production) or missing teats were observed.

### 3.2. Bacterial Species in Normal Canine Milk

The bacterial species identified in the milk samples are presented in Table 1. MALDI-TOF identification and 16S rDNA sequencing of representative isolates supported the species identification and the identification at genus level resulting from the agar culturing, colony characteristics and biochemical testing. Both *S. pseudintermedius* and enterococci were found in the milk of all 11 dams. *S. pseudintermedius* was the most common bacterial species in the milk (75% of positive samples) during early lactation period, followed by enterococci (26%). *S. pseudintermedius* was present in the milk of most mammary gland complexes in each dam, whereas the enterococci were not present to the same extent but still quite common in the separate glands in all the samples. The number of the different bacterial species in the milk did not vary significantly among the analyzed samples.

The mean number of colony-forming units (cfu) of bacteria per mL milk per mammary complex per individual dam is shown in Figure 2. The data represented the mean of the mean of the total number of cfu in 210 samples. The mean of the mean of all sampleswas 16,824 (SEM = 4039) cfu/mL and is indicated in Figure 2 by the straight line. The mean number of cfu/mL per dog varied, but the 1-way Chi-square value of the Wilcoxon/Kruskal–Wallis test was 17.004 and *p* = 0.07, and the observed differences in cfu/mL between the individual dogs were not statistically significant.

There was a statistically significant correlation between the mean number of cfu/mL of milk and the number of days after parturition for the first milk sampling procedure (correlation coefficient = 0.8; *r*^2^ = 0.65; *p* = 0.002). There was also a significant correlation between the mean number of cfu/mL and the number of days after parturition when both sampling procedures were included (correlation coefficient = 0.46; *r*^2^ = 0.22; *p* = 0.03). No significant correlation was found between the mean number of cfu/mL of milk and the age of the dam (correlation coefficient = 0.3; *r*^2^ = 0.09), or the number of puppies born (correlation coefficient = 0.16; *r*^2^ = 0.03). The mean number of cfu/mL per dam varied and the observed differences in cfu/mL between the individual dams were not statistically significant *p* = 0.07.

When comparing the mean number of cfu/mL divided into groups of cfu: >10 to ≤100; >100 to ≤1000; and >10,000, there was no significant differences between left side vs. right side of mammary gland complexes, nor between times of sampling.

## 4. Discussion

The primary finding of this study is that, in early lactation, the milk of postpartum healthy dams with healthy offspring, without clinical signs of mastitis or other illness, contains abundant, culturable bacteria from a range of different species. There was a significant correlation between the number of cfu/mL of milk with an increasing number of days after parturition in both sampling procedures, and high, but not statistically significant, individual variations in cfu/mL among the sampled dams which indicated that the number of cfu/mL of milk grew in every dam as milk production increased. After the evolution of laboratory techniques, different bacterial species in abundance were described in the milk of asymptomatic mammary glands in women, so that modern researchers consider human milk to be an important source of commensal bacteria inhabiting the intestinal tract of newborns, enabling the establishment of healthy bacterial microflora [10,21,22,23]. The origin of the bacterial population in freshly secreted milk is still uncertain. Scientific opinions on this have varied, depending on the species involved. The ascendant route through the orifices of the teat from the neonate’s mouth during sucking, migration of normal bacterial skin inhabitants, shedding of bacteria during lactation and the entero-mammary pathway are some of the potential sources of bacteria in milk and mammary glands [24,25,26,27,28,29,30].

In our study, agar culturing was used as the primary technique. It is not possible to identify all staphylococci, streptococci and other genera to the exact species level by colony characteristics, biochemical diagnostic tools and anamnestic information. Therefore, we selected representative colonies from the milk of two dams selected by the same criteria as the test dams for applying the MALDI-TOF identification and 16S rDNA sequencing. The identification of *S. pseudintermedius* was identical with all three diagnostic approaches and as discussed in Bond and Loeffler [31], *S. pseudintermedius* is the major coagulase positive staphylococcus of the dog, with 20–90% of the dogs being carriers. The coagulase negative staphylococci of dogs often need additional diagnostic tools such as MALDI-TOF or 16S *rDNA* identification to be taxonomically identified at species level. Enterococci as *Enterococcus faecalis* or *Enterococcus faecium* can be identified by culturing and by sugar degradation, but to identify all enterococci to species level, MALDI-TOF or 16S *rDNA* sequencing is required. In this study, only clostridia were found to need an anaerobic atmosphere to grow from milk. The bacteria from healthy milk in this study are the same bacteria that also can be found in many other sites of skin and mucous membranes in dogs and mammals in general from normal microbiota but also from sites with clinical signs of bacterial infection.

Several studies have previously described a microbial population in canine milk, without concluding that it may be a normal canine milk microbial flora [11,13,14,15,32,33]. Two studies evaluated their findings as abnormal [11,14]. The primary goal of these studies was to find a correlation between neonatal death and subclinical mastitis in the dam, and poor milk quality was described as one of the possible causes. In one study it was found a high number of bacteria in canine milk samples, but the authors did not interpret this finding as normal flora and could not exclude potential contamination or mastitis [34]. Another study found no difference in bacterial representation between healthy and dams with mastitis [35]. The most common isolate was *Escherichia coli* followed by *Staphylococcus* sp. *E. coli* is the predominant bacterium associated with mastitis in canines and felines [36]. We, however, found mostly *S. pseudintermedius* followed by enterococci, whereas *E. coli* was present in only one milk sample in this study. Interestingly, this dam was the only one to subsequently develop mastitis at week 4 postpartum. Whether this finding is a coincidence or a correlation can only be speculated. None of her puppies had any clinical signs of illness during the follow-up period. Another interesting finding in a study by Vasiu et al. was that most of the samples from bitches with *Lactatio sine graviditet* were sterile. In pseudocyesis, there is no offspring and produced milk has no purpose to colonize their gut. That could be one of the reasons there was no bacterial growth in most of those samples. This additionally braces our results, which support the existence of naturally occurring milk microbiota that play an important role in forming a healthy bacterial flora for offspring, such as staphylococcal flora (*S. pseudintermedius*), which has been gradually acquired in the mouth of newborn puppies [37] and isolated in 78% of puppies 8 h after birth [38]. We should also mention that enterococci are commonly considered to have probiotic effects and they occur as an important part of the intestinal microbiota of mammals and other animals even if they occasionally can cause various infections in some individuals [39].

A difference between our study and these other canine studies was that we sampled only healthy dams with healthy offspring delivered naturally without the occurrence of any indication of any disorder, septicemia or fading puppy syndrome, so we could minimize the possibility to contaminate the samples. The mode of delivery was important because it has been described that it affects microbiota of newborn puppies [38,40]. We have, however, found more samples with bacterial growth, and a lower number of bacterial species than some of the other studies. A possible explanation could be different culturing techniques applied, since we did not use selective media during the first inoculation, but only as a tool of differentiation between certain bacterial species.

In this study, no significant variations were observed in the bacteriological status of milk in dams of different age. On the other hand, the colostrum of dams under the age of 6 years is reported to be of better immunological quality, i.e., contains a higher number of immunoglobulins, compared to older dams [41]. It is challenging to explain why some of the samples were negative. Possible explanations may be a low sample volume for inoculation, inoculation of unequal fractions of the milk or limitations of the applied culture-dependent methods.

Our results contribute to and strengthen the concept of normal milk as non-sterile fluid that may play a role in populating the normal mammalian intestine with bacteria. A healthy microbiome is crucial for every individual. A recent study in *Theriogenology* reported the existence of bacterial microbiota even in the placenta of newborn puppies, where puppies with bacterial microbiota in placenta gained more weight than puppies born with a sterile placenta [38]. In a similar study, preliminary data were reported after culturing of bacteria from the uterus, amniotic fluid and the meconium of healthy dogs and cats during elective cesarean section at term of pregnancy [42]. How the dam impacts the microbiota of the litter up to two months after delivery was studied by standard culturing techniques combined with a novel approach of analyzing the MALDI-TOF results in another study by the same scientists. They found that the microbiota of the litter over time became less heterogenic during the two first months of life [43]. The impact of the milk from the mother may have been one of the factors contributing to these results. How could this influence the physiological role of bacteria in milk? The colostrum of domesticated mammals contains relatively high concentrations of antibodies, which might also be expected to have a profound influence on milk bacteria. Perhaps these antibodies are selective for, for example, *E. coli*, but not the staphylococci found by us.

We cannot rule out that our milk samples were contaminated during sampling. However, much effort was taken to avoid contamination of the sample from the teat. This included clipping of hair surrounding the teat, disinfection of the teat, discarding the first milk droplets, sampling through a small, thin sterile pipette that did not touch anything but the small droplet on the surface, and a rapid sampling time of one or two seconds. Based on the findings of similar bacteria from left and right side and from all mammary gland complexes, as well as minor differences in the bacterial flora between lactating dams from different households, we strongly believe it is unlikely our findings are due to contamination. Post collection, all the milk samples were kept cool until inoculation on the agar plates, and we also ensured a short time from sampling to inoculation, with a maximum of three h. We believe these precautions would have greatly reduced the possibility of any contaminating bacteria to grow and multiply during transport.

In this study, a culture-dependent method was used, which represents a standard microbiological procedure for the detection of milk bacteria, because we would aim to achieve results relevant to a clinical practice. A typical clinical approach is to regard the occurrence of bacteria in a milk sample as infection-related and to treat the affected glands with antibiotics. In a study of Milani et al., the effect of the use of antibiotic treatment in dams during the prepartum period was evaluated [44]. There was no reduction in the mortality of the puppies but a significant increase in the occurrence of multi-resistant pathogenic bacteria was seen. Considering the global threat posed by antibiotic resistance, such treatments should be avoided when not indicated. Our findings demonstrate that bacterial growth in a milk sample of a healthy dam in the early phase of lactation is normal and does not require antibiotic treatment in absence of clinical signs of infection. We encourage clinicians to sample several milk glands from the same dam, which will enable a better evaluation of the results, instead of only testing a single, suspected gland.

Regarding the limitations of our study, it could be argued that a non-randomized set of samples could introduce bias. Restricting our investigative tools to non-selective culture media and culture-dependent techniques may have caused us to miss some microbes potentially found in milk.

## 5. Conclusions

Milk, just like other body fluids from healthy individuals, was for a long time expected to be sterile. Consequently, any findings of bacteria in a culture of milk were interpreted as infection-related, and clinicians have often treated such “affected” glands with antibiotics. Considering the global threat posed by antibiotic resistance, such treatments should be avoided when not supported by other findings of inflammation. In this study, consistent populations of bacteria in milk samples taken from normal, healthy dams with normal, healthy puppies were found. These results suggest that they should be considered as components of the normal canine milk microbiota, instead of mere contaminant bacteria. In spite of the low sample size, the results of this study are not negligible and could serve as a good starting point for future studies related to milk and the microbiome of puppies and their interconnection.

## Figures and Tables

**Figure 1 animals-13-02206-f001:**
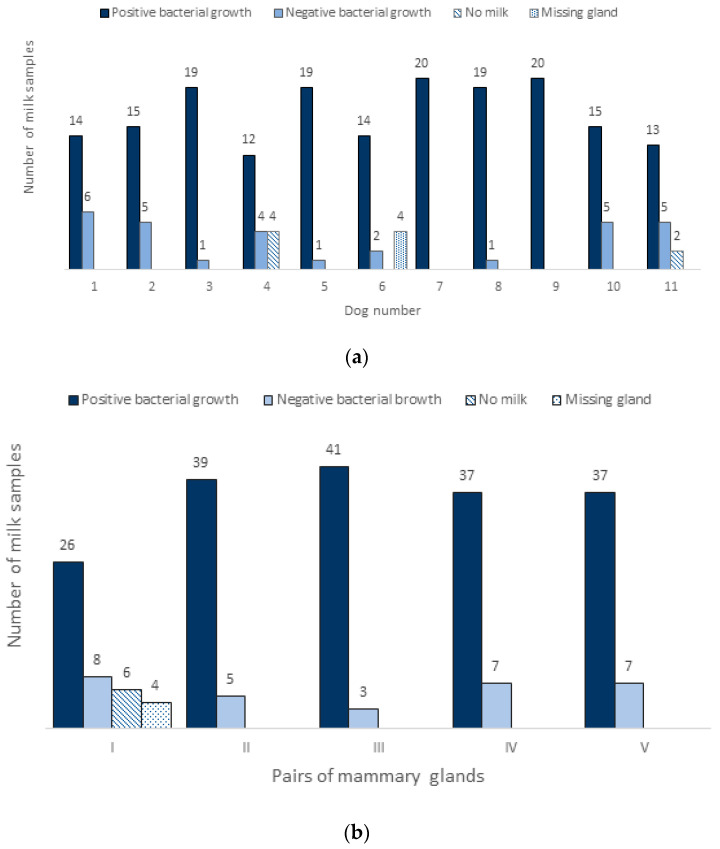
(**a**) Presentation of bacterial growth from milk samples during both postpartum sampling procedures. Samples obtained from all the dams and their teats separately (max. 20 samples per dog; 10 teats × 2 times sampling); (**b**) Bacteriologically positive and negative samples in normal lactating dams clustered in five groups from I to V (I and II = cranial and caudal thoracic, III and IV = cranial and caudal abdominal, and V = inguinal mammary pairs). Each dam was sampled twice, one week apart, and both results are included in the figure (one pair = 2 glands on both sizes × 2 sampling procedures × 11 dogs = 44 samples per pair). The first pairs yielded the lowest bacterial growth.

**Figure 2 animals-13-02206-f002:**
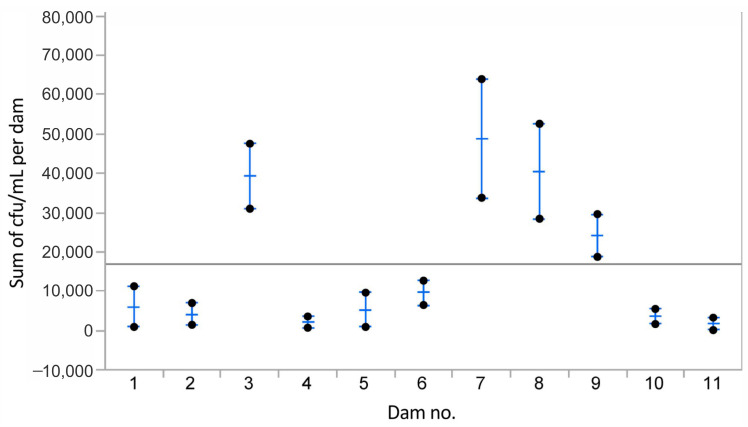
The mean numbers (and SEM) of colony-forming units (cfu) of identified bacterial species per mL of milk from the mammary gland complexes of all dams. The horizontal grey line represents μ $\bar{x}$ value.

**Table 1 animals-13-02206-t001:** Distribution of bacterial species identified in milk samples of healthy dams using culture-dependent method for aerobic and anaerobic cultivation. Results from both sampling procedures are included.

Bacterial Species	Dam 1	Dam 2	Dam 3	Dam 4	Dam 5	Dam 6	Dam 7	Dam 8	Dam 9	Dam 10	Dam 11
*Staphylococcus pseudintermedius*	+	+	+	+	+	+	+	+	+	+	+
*Enterococcus* spp.	+	+	+	+	+	+	+	+	+	+	+
*Clostridium* spp.		+		+	+	+	+	+	+	+	+
Coagulase negative staphylococci	+		+	+	+	+		+	+		
*Streptococcus* spp.	+	+	+			+			+		
*Streptococcus canis*			+						+		
*Bacillus* spp.			+								
*Pasteurella* spp.		+									
*Escherichia coli*											+

## Data Availability

The datasets used and/or analyzed during the current study are available from the corresponding author on request.

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
