# Peer review of "Bacteria in Normal Canine Milk Analyzed by Blood Agar Medium"

_animals, 2023, doi:10.3390/ani13132206_

Round 1

Reviewer 1 Report

I have loved to read this paper which is of a very good level, both in its presentation and in the value of its content. I have really very few  suggestions which are mostly curiosities for authors together with my congratulations for the very good work done.

- Line 30-31: why enterococci is written in this way, while all the others are written in italics and as sp.? Was it wrong to write Enterococcus spp. in italics?

- line 50: This healthy microbiota has antimicrobial activities: how this is going to happen?

- line 61-63: It is just my personal opinion but... I really don't find useful to use open questions marks in the context of an introduction where it is described the aim of the work. I feel like it is more straightforward to write directly the hypothesis of the work. 

- line 75-76. Did you consider just the antimicrobial treatment as exclusion criteria? 

- line 100-101. All teats were disinfected using sterile compresses soaked in 70% ethanol : no time of exposure to ethanol was considered? 

- line 132-134. I am very interested in this MALDI_TOF technique of identification. May you spend some few words in explaining the method and /or adding any reference of that? 

- line 152: Same statistics were calculated for the dams combined.  It means an overall? 

- line 211-212: The data represented the mean of the mean of total number of cfu/mL each dam of all the mammary gland complexes, and of left and right side, and both sampling procedures. this sentence is not totally clear to me, please may you rephrase? 

- line 261: such as

- line 264: by sugar degradation

- In the discussion, two recent papers should be added as they are reporting similar results of  your beautiful work, and may enforce the hypothesis of the authors. the works are :

Banchi, P. et al. Challenging the Hypothesis of in Utero Microbiota Acquisition in Healthy Canine and Feline Pregnancies at Term: Preliminary Data. Veterinary Sciences, 2023, 10(5), 331

Del Carro A., et al. The evolution of dam-litter microbial flora from birth to 60 days of age. BMC Veterinary research, 2022 18(1), 95

Moreover, an old work referring to some threatening effects on the microbial flora due to excessive use of antibiotics on breeding bitches is reported here (with some findings on the mammary milk secretions which are somehow reinforcing yours): 

Milani C. et al., Antimicrobial resistance in bacteria from breeding dogs housed in kennels with differing neonatal mortality and use of antibiotics. Theriogenology 2012, 78:1321-1328

Author Response

Point-by-point response to Reviewer 1: 

Line 30-31: It is not wrong to use the designation "enterococci" without italics. However, to be more consistent with the other naming in this sentence we believe it is more appropriate to use the designation "Enterococcus spp.". Table 1 was also adjusted accordingly.

Line 50: To indicate what antimicrobial activities that can come from an "healthy" microbiota we have made an addition to the sentence: "This healthy microbiota has antimicrobial activities such as production of bacteriocins or other antimicrobial products like organic acids and may reduce.. 

Line 61-63: We agree with the suggestion of avoiding question marks in the introduction related to the aims of the work. We have adjusted the sentence minimally to avoid the question mark. 

Line 75-76. We wanted to exclude individuals with any recent antibiotic treatment to not affect the microflora, which led us to use 2 weeks without antibiotic usage as a treshold for inclusion of the dams into the study. We also could not gain secure information about the lifelong history of antibiotic usage for each dog.

Line 100-101. All teats were disinfected using sterile compresses soaked in 70% ethanol but we waited until the ethanol was evaporated before we started sampling of milk. The time before the skin was dry varied from less than half a minute to a couple of minutes do different physical conditions. We have adjusted the text and a parenthesis with " approximately 30 seconds".

Line 132-134. It is a good suggestion to add a reference for the MALDI-TOF method. We added the reference: 

Dingle TC, Butler-Wu SM: MALDI-TOF mass spectrometry for microorganism identification. Clinics in laboratory medicine 2013, 33(3):589-609.

Line 152: The same statistics were calculated for the dams combined was employed and it means an overall calculation for the dams as unit.

Line 211-212: The sentence: "The data represented the mean of the mean of total number of cfu/mL each dam of all the mammary gland complexes, and of left and right side, and both sampling procedures." is long and complex and we have adjusted the text by splitting up and rephrasing to make it more simple and clear. 

Line 261: The word "such" has been added in front of "as MALDI-TOF"

Line 264: The word "by" was added before "sugar degradation"

In the Discussion we have added all three suggested references with some extra text. The suggestions for additional references strengthened the Discussion and linked the study better to the existing results in the same scientific field of research.

 Thanks a lot for the suggestions and comments. Those have improved the manuscript and linked the study more optimal to previous research in this field.

Reviewer 2 Report

The aim of the present study was to investigate whether there is a normal, culturable bacterial microflora in the milk of healthy dams during early lactation that causes no health issues to the puppies, and if so, to describe the bacterial species that are present and to estimate the populations of bacteria in milk using standard culture-dependent techniques for detection of bacteria.

Research is interesting as well as it has a scientific value. The introduction provides a good, generalized background of the topic that quickly gives the reader   appreciation of the scientific relevance and timeliness of the research theme. I think the findings of this study are sufficiently described in the context of the published literature.  The conclusions are supported by appropriate evidence. 

Particular attention should be paid to the fact that this study has been carried out, to a rich set of analyzes and to the high application potential of the obtained results.

I have some specific comments on the manuscript:

  • Figure 1 isn’t readable. It needs to be necessarily improved,
  • Line 65: please, replace word “flora” with “microflora”.
  • Please, write the abbreviation “cfu” in the whole text of manuscript in a uniform way (Figure 2: title and Y axis).

Overall it  is a well-written article with a significant application potential.

From my standpoint, this manuscript is appropriate for publication in Journal – Animals,  after minor revision given the above aspects.  

Author Response

Point-by-point response to the Review Report (Reviewer 2):

Thanks for the insightful comments to the study presented.

We agree that Figure 1 was having too small characters to be easily read. We have split the two parts of Figure 1. Part (a) of the figure is given the full width of the side and similarly we have placed Part (b) below Part (a). By doing that Figure 1 will be easy to read and overlook.

Line 65: We have replaced the word “flora” with “microflora”. Thanks for reminding us.

Regarding the consistent use of "cfu" throughout the manuscript we have added a parenthesis in the text of Figure 2 explisitly explaining the designation on the Y-axis which is important to understand the data in presented in Figure 2.